# Cancer Survivors’ Disability Experiences and Identities: A Qualitative Exploration to Advance Cancer Equity

**DOI:** 10.3390/ijerph19053112

**Published:** 2022-03-06

**Authors:** Susan Magasi, Hilary K. Marshall, Cassandra Winters, David Victorson

**Affiliations:** 1Department of Occupational Therapy, University of Illinois at Chicago, 1919 W. Tailor St., Chicago, IL 60612, USA; hmarsh3@uic.edu; 2Department of Disability and Human Development, University of Illinois at Chicago, 1640 W. Roosevelt Rd., Chicago, IL 60612, USA; wintersc@uic.edu; 3Department of Medical Social Sciences, Northwestern University Feinberg School of Medicine, Chicago, IL 60611, USA; d-victorson@northwestern.edu

**Keywords:** cancer survivorship, qualitative methods, persons with disabilities, identity, cancer equity, ableism, stigma

## Abstract

Eliminating cancer-related disparities is a global public health priority. Approximately 40% of cancer survivors experience long-term effects of cancer which can lead to activity limitations and participation restrictions; yet discussions of disability are largely absent from clinical and research cancer health equity agendas. The purpose of this study was to explore how cancer survivors experience and make sense of the long-term disabling effects of cancer and its treatments. In this qualitative study, data were collected via in-depth semi-structured interviews with survivors of breast cancer, head and neck cancer, and sarcoma (*n* = 30). Data were analyzed thematically using a 2-phase iterative process proceeding from descriptive to conceptual coding. Survivors experienced a wide range of long-term physical, sensory, cognitive, and emotional effects, that intertwined to restrict their participation in self-care, work, leisure, and social roles. While the interaction between impairments and participation restrictions meets the definition of disability; participants articulated a range of responses when asked about their disability identity, including (1) rejecting, (2) othering, (3) acknowledging, and (4) affirming. Findings may be indicative of structural and internalized ableism which can impede cancer care and survivorship. To support cancer survivors’ transition to post-treatment life, cancer care providers should implement anti-ableist practices and engage in frank discussions about cancer’s long-term impacts.

## 1. Introduction

The elimination of cancer inequities is a global health priority. Social groups which have historically experienced discrimination, exclusion, and marginalization are at risk for cancer inequities [1]. The cancer health equity agenda is an umbrella term used to describe a wide array governmental, non-governmental, and clinical initiatives, programs, and policies aimed at reducing the disproportionate burden of cancer experienced by certain groups, especially those from racial and ethnic minority communities. People with disabilities (both those with pre-existing disabilities and those who acquire disabling conditions as a result of cancer) are the largest minority group in the United States and are largely absent from the cancer equity research, clinical, and public health agendas [2]. Biopsychosocial frameworks recognize disability as being socially constructed when people with bodies and minds that function outside the norm encounter barriers in the physical, social, economic, and political environments [3]. The disability experience is strongly influenced by the normalization of ableist attitudes and social structures. Ableism is stereotyping, prejudice, discrimination, and social oppression of people with disabilities [4]. Structural ableism is embedded in social structures, including the health and cancer care systems [5,6,7,8]. People with disabilities and chronic health conditions, such as cancer, may unconsciously internalize ableist beliefs which can compromise their quality of life and access to healthcare services [9,10].

An estimated 16.9 million cancer survivors live in the United States [11,12]. Approximately 40% of survivors experience long-term physical, cognitive, and psychological effects [13] due to cancer and its treatments (i.e., issues that develop during active treatment and persist for at least five years after the completion of initial cancer treatment) [3]. Common long-term sequelae include pain, fatigue, cognitive effects (e.g., chemo-brain), and psychosocial distress, such as anxiety and depression; which can in turn lead to activity limitations and participation restrictions. These long-term effects can negatively impact social participation and health-related quality of life [14]. Yet, cancer survivors report that these issues are inadequately addressed within the cancer care system, leaving patients to navigate the impact and long-term management of these symptoms on their own [15].

The World Health Organization’s International Classification of Functioning, Disability and Health (ICF) is a biopsychosocial framework that has been used to understand the proximal and distal long-term effects of cancer and its treatment [16,17]. The ICF is a useful heuristic to understand how these long-term effects contribute to significant disability among cancer survivors. The ICF recognizes the dynamic interplay between health conditions (such as cancer and its treatments) with bodily symptoms and side effects (such as pain, fatigue, and mental functions), which in turn contribute to more distal outcomes such as activity limitations and participation restrictions. The ICF offers a broad lens to understand disability, as it acknowledges both the need for curative and rehabilitative approaches to prevent and treat disease and dysfunction, while addressing the equally important goal of maximizing participation in meaningful life activities [18].

Disability rates vary by cancer type, cancer stage, and treatment protocol. For this study, we identified three common cancers whose survivors often experience long-term disability: breast cancers [19,20], sarcomas, especially when the treatment protocol includes amputation [21,22,23] and head and neck cancers, especially when the treatment protocol includes neck dissection [24]. Common long-term negative effects in these three groups of cancer survivors include: (1) pain [23,25,26,27], (2) fatigue and disturbed sleep [28,29,30], and (3) psychosocial distress, including anxiety, depression and altered body image [31,32,33].

Any discussion of disability among cancer survivors must acknowledge two important realities: (1) the heterogeneity of the disability experience and (2) the contentious nature of the term ‘disability’. There is tremendous diversity in the nature and severity of individual impairments and their impact on people’s day-to-day experiences. The disabling long-term effects tend to be both under-diagnosed and difficult to treat [34]. Cancer survivors often feel unprepared and unsupported when dealing with long-term disabling symptoms and their impact on social roles and participation [34]. Many cancer survivors with disabilities experience negative social outcomes as a result of barriers in the physical, social, economic, and political environment. As a result of external and internalized negative biases against disability, many cancer survivors eschew the label of disability [35,36]. However, acknowledging one’s disability status may help increase self-worth, self-efficacy and access to available health and community-based services as well as social supports [37].

As part of a larger study aimed at developing an mHealth self-management intervention for people with disabilities navigating the long-term effects of cancer and its treatments, we conducted formative semi-structured qualitative interviews with cancer survivors (*n* = 30). Based in part on the recognition that an intervention is only useful if it is getting to the people who need it, we sought to understand how people make sense of their post-treatment experiences. Specifically, while the research team’s conceptual grounding in the ICF may recognize the long-term effects of cancer as contributing to disability status, we did not want to alienate our target audience by imposing a potentially stigmatized identity marker upon them. Therefore, the purpose of this qualitative interview-based study was to examine how survivors of breast cancer, head and neck cancer, and sarcoma negotiate disability understandings and experiences when dealing with the long-term effects of cancer and its treatment.

## 2. Materials and Methods

### 2.1. Study Design and Research Paradigm

This cross-sectional exploratory qualitative study was performed using reflexive thematic analysis [38] within a social constructivist paradigm. The guiding principles of social constructivism are that knowledge is created through human interactions and that people make sense of their world through recursive processes, experiences, and interpretation [39]. Social constructivism is particularly useful for examining ambiguous and evolving concepts and experiences, such as the intersection of long-term survivorship and disability identity.

Qualitative findings are reported in accordance with standard reporting guidelines for qualitative research [40].

### 2.2. Research Team and Positionality

Our inter-disciplinary team includes doctorally trained disability and cancer researchers, clinical health psychologists, occupational therapists, computer scientists, and an anthropologist, all of whom have extensive experience and training in qualitative research with diverse populations, including cancer survivors and people with disabilities. Of note is the lead author’s clinical and research expertise in occupational therapy and disability studies which informed the study’s focus on social participation and disability identification.

### 2.3. Participants

In collaboration with cancer support community-based organizations and two academic cancer centers in a large urban center in the Midwestern United States, we conducted community outreach and disseminated recruitment flyers to cancer survivorship networks. Inclusion criteria were adults over the age of 18 years with a self-identified history of breast cancer, head and neck cancer or sarcoma who self-identified as having a disability. Interested individuals were asked to contact the research team to learn about the study. Upon contact, the study purpose and protocols of the study were explained and interested individuals were screened for eligibility. Disability status was verified using a 2-step screening process. First, interested individuals were asked if they considered themselves to be a person with a disability. For the purposes of this study, we adopted the Americans with Disabilities Act’s broadly inclusive definition of disability. The American with Disabilities Act (ADA) states: “The term ‘disability’ means, with respect to an individual, (A) a physical or mental impairment that substantially limits one or more of the major life activities of such individual; (B) a record of such impairment; or (C) being regarded as having such an impairment”. If participants responded “yes” to the screening question, they were then asked the standard 6-disability questions from the American Community Survey. Individuals were excluded if they were unable to provide informed consent, participate in a 90-min Zoom-based data collection session, or communicate in English.

Eligible participants then completed the informed consent process and provided signed informed consent via the secure Redcap Online system.

### 2.4. Setting

Due to constraints imposed by the COVID-crisis, all data were collected remotely using a secure Zoom connection. The cancer survivors were asked to select a quiet, private, and comfortable location within their home for the interviews. The research team provided technology assistance as needed to support participation.

### 2.5. Data Collection

The interview guide was developed through an iterative collaborative process with the interdisciplinary research team. The interview guide was structured to address the broader study’s aims of developing a self-management intervention and included key conceptual categories of (1) cancer impact, (2) self-management strategies, (3) disability identity, and (4) health information technology use and preferences. Duration of interview sessions ranged from 25–105 min. All interviews were conducted between May and July of 2021 by a research team member with clinical training as an occupational therapist and experience in qualitative methods, disability, and cancer rehabilitation (SM, CW). Participants had no established relationship to the interviewer before data collection.

Interview sessions were audio-recorded with the participation’s permission. Auto-generated transcripts of the Zoom recording were verified and corrected by a member of the research team to ensure that the data were of high quality prior to importing them into the qualitative data management and analysis platform, Dedoose, for team-based analysis. Upon completion of the data collection session, the interviewer wrote a structured fieldnote to synthesize and summarize the session and to track data saturation, a measure of qualitative rigor and technique used to determine the adequacy of the sample [41,42]. Saturation of broad thematic areas was firmly established after 21 interviews, however, the research team continued participant enrollment for pragmatic and conceptual reasons. Pragmatic reasons were based on participants’ assertions that they were rarely asked about the long-term effects of their cancer, and they valued the opportunity to describe those experiences so that others may learn from them. Therefore, given the minimal risk and burden to participants, we continued enrollment and data collection until all interested and eligible individuals from our community outreach efforts were interviewed. Conceptually, from a social constructivist and reflexive thematic analysis perspective, the notion of saturation is contested as people make and re-make meaning of fluid identity markers such as survivorship and disability [43]. Therefore, additional data collection beyond thematic saturations added depth and nuance to the understanding of cancer survivors’ disability experiences and identities. To minimize burden on participants experiencing the long-term disabling effects of cancer and its treatments, transcripts were not returned to participants for member-checking. Instead, we employed a process of peer examination as recommended by Creswell and Miller (2000) to support the validity of our findings [44]. Specifically, synthesized findings were distributed to four cancer survivors with long-term disabilities and two disability studies scholars with clinical backgrounds in occupational therapy and cancer care for review and critical feedback [44]. Several participants asked to be apprised of the study findings and these requests are being honored.

### 2.6. Data Analysis

A 2-person data coding team (Susan Magasi, Hilary K. Marshall) engaged in an iterative and collaborative coding and reflexive thematic analysis process proceeding from descriptive to conceptual coding through an iterative and recursive analytic process [45]. Specifically, both coders independently read a sub-set of transcripts and then convened to define a preliminary coding dictionary based on the targeted research question. Once the preliminary coding dictionary and tree were defined and triangulated, HKM completed the descriptive coding of the full dataset in Dedoose, adding additional codes as needed to reflect the emergent data. The team then convened regularly to discuss impressions and refine the coding dictionary. The preliminary coding tree was informed by the conceptual categories that framed data collection, but refined and revised based the emergent data. Once the entire dataset was coded, the lead author organized the data into meaningful conceptual categories to address the targeted research purpose of understanding how cancer survivors negotiate disability understandings and experiences when dealing with the long-term effects of cancer and its treatment. Interpretations were discussed with the research team and refined by co-authors.

## 3. Results

Our final sample included 30 cancer survivors. The sample included proportionally greater numbers of women breast cancer survivors compared to head and neck cancers and sarcoma. When examining the collective incidence of these three cancer types in the United States, sarcoma represents roughly 4%, head and neck roughly 19% and breast cancer roughly 78% of these three groups combined [46]. Based on this, we enrolled over 4 times more sarcoma patients than the national incidence (17% of our sample); roughly the same proportion of head and neck patients based on national incidence (20% of our sample), and slightly fewer breast cancer survivors than national incidence estimates (67% in our sample). Our quotations also reflect these percentages. Participants ranged in age from 22–79 years. One participant had a pre-existing disability. Three participants had more than one separate cancer diagnoses not due to metastases. Length of time post-treatment varied greatly from 19 years to less than one year. Table 1 provides brief demographic information about each participant.

In this analysis, we examined disability from two perspectives. First, we drew on definitions of disability that identify a person with a disability as someone with a physical or mental impairment that substantially limits a major life activity by examining how long-term effects intertwined to impact survivors’ participation in meaningful life activities, including self-care, work, leisure, and social relationships. Second, we examined participants’ disability identity. Although inclusion criteria were based on established criteria for defining disability, upon direct questioning during the qualitative interview, half of the participants (15/30) stated they did not consider themselves to be disabled. Indeed, we identified a spectrum of responses that ranged from (1) rejecting, (2) othering, (3) acknowledging, and (4) affirming. We will explore each of these themes with supporting data.

### 3.1. Long-Term Effects

Consistent with the previous literature, the participants described a wide range of symptoms and long-term effects from cancer and its treatment that affected their bodies, minds, thoughts, and feelings. Table 2 provides an overview of the long-term impacts experienced by participants broken down by functional area (body, brain, and thoughts and feelings) along with representative quotations. Some symptoms, such as fatigue, cognitive fuzziness, and anxiety, were generalized and experienced by survivors across all three diagnoses. Other symptoms were more cancer-specific, such as dysphagia and swallowing difficulties among head and neck cancer survivors. It is important to recognize that symptoms rarely occurred in isolation but rather participants reported dealing with a variety of issues across multiple body systems. Survivor 18 spoke to the challenge of dealing with the complexities of post-treatment life.
*The only thing worse than going through cancer treatment is after cancer treatment…that first month or two after treatment was like oh my gosh, you’re relearning your whole life…it’s a whole new mindset. All your dreams, everything you ever thought, is different now.**(Survivor 18, sarcoma)*

The representative quotations in Table 1 reinforce the understanding that the impact of long-term effects had a profound influence on people’s post-treatment lives and sense of self.

### 3.2. Lack of Preparedness

While these symptoms and long-terms effects are well-documented in the clinical and research literature, many participants indicated that they were unprepared for the severity and impact that cancer would continue to have on their lives post-treatment.


*The long-term negative effects, people don’t talk about. You hear all this “I’m a survivor and I feel great after!” You know like I don’t feel great… I would hate to sound like hopeless but I’m sure there are people who would benefit from knowing that they’re not the only one [for] who it wasn’t just a bump in the road”.*

*(Survivor 22, breast cancer)*


The unpredictability around long-term impacts made it particularly challenging for people to reconcile their experiences with their expectations of survivorship.


*The issue is that, well what else am I going to look forward to happening to me? What other kind of miserable thing is going to happen to me? They don’t tell you all that stuff. You know? You can get a list of stuff from cancer, but it could be days, months, years. You don’t know what’s going on, what’s coming down the line. When they say long term side effects, they’re not kidding, but it’s glossed over because it hasn’t happened.*

*(Survivor 12, head and neck cancer)*


Adding to the uncertainty of post-treatment life was the evolving nature of some long-term effects that caused several participants to question why they had fought so hard to cure their cancer in the first place.


*After I was done with the chemo, done with everything else. I felt like every six months, three to six months after, I was diagnosed with another problem. I felt like wait a minute, they told me I was all good. They told me life was going to be great. Things are gonna be good. Now they’re telling me ‘you’re getting sicker’… What? After that, it was like I beat cancer for this?*

*(Survivor 24, breast cancer)*


Lack of preparedness is further exacerbated by pollyannaish media portrayals that leave no space for the unpleasant realities of persistent symptoms post-treatment.


*You see all these happy people on commercials and you know, you think Yeah, I’m just gonna get treatment, I’ll be done. That’s not my case… They don’t tell you it is going to be permanent. I remember the moment I realized this neuropathy is not going away. I was a puddle on the floor, you know. [But] you pick yourself up and get over it. You manage it, but I understand, my God, nobody would go through cancer treatment if they know all the ‘what ifs”.*

*(Survivor 23, breast cancer)*


### 3.3. Participation Restrictions

One of the big ‘what ifs’ was how various long-term effects on people’s bodies, minds, thoughts and feelings intertwined to impact their role performance and social participation. These changes were both significant and insidious. Survivor 15 spoke of the ways that physical impairments have crept into her life and started restricting more of her activities of daily living.


*I knew something was wrong with my arm when I couldn’t fasten my bras and I’ve always been able to even after surgery, I’ve always been able to fasten them in the back… [Then] I could not even get the arm back, and that’s why showering is even difficult because I can’t do the back scrub even though I have a brush sponger kind of thing. Not being able to fully use the arm to an extent. Today I had to stop by the ATM. I literally had to get out of the car because the ATM is on the left arm side… So it limits my ability to do things.*

*(Survivor 15, breast cancer)*


Numerous participants spoke of the permanence and prolonged impact of their long-term effects. For example, survivor 20 described how she is going to have to deal with the realities of lymphedema for the rest of her life.


*I am supposed to wear a sleeve and a glove for the rest of my life… My profession prior, I was lifting cots and things like that, very reliant on my left arm. Even trying to hold my great nephew, who’s just turned nine months. Two months ago, my niece in-law, I was holding him with my left arm for his head, with a bottle in my right… And she said his head had to go back further and my arm wouldn’t go back further. And she’s not as understanding as she could be… She just grabbed him because my arm didn’t go back further. And that hurt.*

*[Crying] (Survivor 20, breast cancer)*


This passage highlights the profound impact that physical symptoms, such as lymphedema and decreased upper body range of motion, have on work and social roles as well as the emotional toll of not meeting other people’s expectation. Survivor 13 also described how her physical limitations impacted her leisure and social roles and opened her up to social judgements from her peers.


*It was my best friend’s birthday… She really wanted to rent bikes and have all of us ride bikes… I just wasn’t capable of doing that with my neuropathy… Even if I pushed myself, if I did it, I would just have to pop Vicodin all the next day… So I just like took an uber to the next place… Then she wanted to go roller skating and like, you know, I played the games in the arcade like I wasn’t going to be a spoilsport, but like I couldn’t be a part of a couple of hours of activities… It was just disappointing; It also just felt kind of awkward, like they were all having these, like, bonding, bonding experiences. Yeah, I just kind of couldn’t… It was kind of a double bind because either I disclose my status to those people who I didn’t know as a cancer survivor and explain why I couldn’t do it or I just don’t do it. *

*(Participant 13, breast cancer)*


In addition to social exclusion from group activities, her inability to engage in leisure roles opened up thorny issues around disclosing her status as a cancer survivor. As survivor 20′s story of holding her great nephew revealed, even when people do disclose their status as a cancer survivor, there is no guarantee that these claims will be accepted, respected, and accommodated. This was particularly common in work situations and several participants were forced to change jobs, quit working or retire early. Decisions about work were influenced by a variety of factors, including the severity of their symptoms, the nature of their work, and their age.


*I just found my concentration is not what it used to be… I ended up taking early retirement last year and I had planned to work a few more years. But I was struggling and people…were just like not treating me well. They made comments… Someone in H.R. even said to me ‘How do we know you really have cancer?’... Because people look at me and they’re like you look ok. Well, just because I look ok doesn’t mean I’m not struggling.*

*(Survivor 9, head and neck cancer)*


These quotations illustrate how stigma around people’s impairments and functional limitations contributes to the loss of meaningful life roles and limits avenues for social participation, which in turn influenced survivors’ sense of self, their relationships with other people, and their financial stability. These factors had a cascading effect on overall quality of life and life satisfaction as captured in this reflection from survivor 24.


*What I find is I started cutting things out of my life and I didn’t realize that I was shortening my life or losing things. You just start adapting to stop doing things that you just can’t do and you realize you’re not living life.*

*(Survivor 24, breast cancer)*


### 3.4. Disability Identity

According to contemporary definitions of disability, these dynamic interactions between bodily impairments and the contextual task demands that result in participation restrictions that many of the survivors in our sample experienced would objectively be classified as a disability. Yet, when asked directly if they considered themselves to be a person with a disability, half of the participants denied the disability identifier. Indeed, participants had a broad range of responses both when denying and claiming disability, including (1) rejection, (2) othering, (3) acknowledging, and (4) affirming.

#### 3.4.1. Rejecting

Survivors who rejected disability identifiers (in spite of significant impairment effects) often had a strong emotional reaction to the term and/or to being categorized with people who they deemed to be less capable and competent than they were.


*I was applying for jobs… When I saw the list of disabilities that I had to check and cancer was on there, I was so mad. Because now I’m considered a disabled person, because I have this? I thought that was ridiculous… I was really mad and I don’t know when that became—when did cancer become a disability?... Maybe I’m not understanding what the true definition of disabled means, but I don’t feel that I’m a disabled person.*

*(Survivor 25, breast cancer)*


Similarly, some participants adopted medical model thinking which conceptualizes disability as a problem that needs to be eradicated through intervention. Just as the cancer had to be “fought,” so too did people’s perceptions of disability, as seen in the following quotation.


*I fight against all of what other people might consider my disabilities, like the neuropathy in my feet, that I get acupuncture for. The talk therapy for my cancer survivorship and depression. My hypnosis, chanting, meditating, walking three to five miles a day. I do all kinds of things to keep myself at my best level that I’m able to achieve. But I am diminished, but I am not disabled.*

*(Survivor 11. Breast cancer)*


The invisible nature of many of cancer’s long-term impacts made it difficult for survivors to claim disability while opening them up to negative judgements based on people’s interpretations of their inability to fulfill normative social roles and responsibilities. For example, when asked if she identified as a person with a disability, survivor 24 eschewed the label, but acknowledged the social stigma she experienced.


*No, not really, because when people see you like that, then they go, ‘What’s wrong with you?’… I’m not missing a leg. And they look at you go, ‘There’s nothing wrong with her”. How do you explain my brain doesn’t work? I have aches and pains that you can’t see. They don’t get that. And then when you sit on a couch all day, they go, ‘Oh, she’s just so lazy”. They just don’t get it.*

*(Survivor 24, breast cancer)*


Unfortunately for participants, denying the label of disability did not change the impact of their impairment on their day-to-day lives. Asserting other aspects of their identity was another strategy that people used to distance themselves from disability.


*There was one week I lit two fires in my shop and didn’t realize until I saw the flames that I had done that. I didn’t smell anything burn. So there is a disability. But when I think of the label for myself, you know, man, husband, father, mechanic, I don’t think of disability as a label that I am sure makes sense.*

*(Survivor 4, head and neck cancer)*


By claiming his multiple roles and identities, survivor 4 was rejecting the essentialism that frames disability as a master status that diminishes and dehumanizes the individual. Similar efforts to reject the stigma of disability identification were seen amongst a small group of participants who couched their rejection in belief in the power of positive and negative thinking and that by using the term they were manifesting a future for themselves that was contrary to their hopes and dreams. For example, when survivor 30 was asked if she considered herself to be a person with a disability, she asserted:


*I don’t. There are some things that I would like to be able to do more that I feel like I can’t do. But like the physical limitations, I try not to let that affect my view of myself overall. I feel like I’m more than just body parts or daily duties. And so I think mindset is really important. And for me, a mindset that’s like positive and even maybe inspiring in some way is not that I feel inspired about myself all the time, but to strive towards that.*

*(Survivor 30, breast cancer)*


#### 3.4.2. Othering

A second strategy that people used when denying a disability identity was to establish a hierarchy in which they admitted to experiencing impairments, but asserted that they did not reach some arbitrary threshold that would move them into “the other” group of people with disabilities. Survivor 23 offered an interesting example of othering as she negotiated the boundaries between her experiences as a cancer survivor and disability.


*The reason I don’t want to say I’m disabled is because I don’t want to take a space for somebody that is really disabled. I don’t have a problem with the term. I mean the term may be a little wonky but if I was disabled, I would own it. I don’t have an ego problem with it. I just don’t want you guys to think I’m disabled. I have been compromised in some ways because of my cancer treatment. But I am by no means what I would define the word disabled as being. I don’t need special consideration… For me disability is too strong a word... There is a continuum in my mind between am I compromised or am I disabled? I would say I’m compromised; I’m not disabled. Everybody you talk to is going to have different experience so maybe that’s the continuum. Do you move from compromised to disabled or where do you fall and what resources do you need?”*

*(Survivor 23, breast cancer)*


In laying out the distinctions between being compromised or being disabled, survivor 23 was couching her denial of disability in principles of benevolence and fairness. Survivor 6 similarly acknowledged disability discrimination, but sought to differentiate himself from people holding those prejudicial views.


*I would say that I’m a person that’s had some of their functionality impaired, but not disabled… Disability came out of handicapped, handicapped came out of cap in hand, where you would sit in a gutter and you’d hold out your cap begging for alms. And so it’s a very, very negative tone. Disability is, by those who are not disabled, seen to be a negative thing. I think disability to me, tells me that if I see you and you have the sticker on your car, that tells me you’ve got to work harder to get through your day than I do. So I respect you… I think the person’s got more guts than the average guy or person.*

*(Survivor 6, head and neck cancer)*


In this passage, survivor 6 was working out his attitudes around disability in real time by first acknowledging ableist attitudes towards people with disabilities. He then swings the pendulum from historically negative views of disabled people as takers to a more benevolent form of ableism that puts the person with a disability on a pedestal, all the while distancing himself from identification with the group. Survivor 27 employed a similarly benevolent framing to position the disabled other as both a source of pity and gratitude.


*The benefit of the thought that you are disabled is in the recognition that your disability is not as profound as another person’s disability and that you need to be grateful. My mother used to have this saying “God bless the mark”. And that would be a declaration of two things. For her, it was a call to pray for that person to receive help and to recognize that you were blessed with the capability that that person can’t do.*

*(Survivor 27, breast cancer)*


While survivor 27 acknowledged a disability status, she used a strategy of othering to separate herself from the aspects of disability that might be seen as diminishing. In this way, we see that people may use multiple strategies to make sense of their disability experience and that the boundaries between categories are porous.

#### 3.4.3. Acknowledging

Many of the participants who did acknowledge themselves as a person with a disability did so grudgingly after a period of trying to make sense of the changes that cancer and its treatments wrought on their lives and sense of self. For example, survivor 1 described a lengthy process of trying to reconcile her new way of being in the world with her self-identity before cancer.


*It has taken me two years to admit it… I’m disabled. I’m disabled because of people staring at me. You know they are staring at me for the wrong reason now. Not because I have a great outfit on. They’re staring because I’ve got a silver leg… I didn’t really accept that I was disabled at first… It’s when you are out in public that you have to say to yourself, ‘Wow. I’m disabled”.*

*(Survivor 1, sarcoma)*


Additionally, inherent in her discussion was the negotiation between the public and private self and how confronting the attitudes of others forced her to acknowledge her status as a person with a disability. Similarly, some survivors acknowledged disability but felt diminished by it.


*I don’t enjoy being disabled. I don’t enjoy that I can’t have a job. I don’t enjoy that, you know—it’s like I always have to be the special person out and it’s not because I’m super talented and great… It’s because I can’t do it. And it’s very annoying. And it’s very depressing… But you know then on the other side, the cancer, people are like ‘Hey, you’re alive”. I’m like, yes but am I quality of life-ing?*

*(Survivor 8, breast cancer)*


Several of the participants reported the strategic use of disability identifiers to qualify for services and supports. For example, survivor 14 discussed the importance of choosing how to self-identify and when to self-disclose one’s disability status while simultaneously acknowledging claiming disability as a way to assert one’s rights to legal protections under federal civil rights laws.


*Disability, if you have to bring… somebody’s attention to something that you’re entitled to by law… it’s your choice how you want to interpret disability. But by law you are defined as disabled and if you need to use it, use the term. Like, you know, ‘I, I’m entitled to an accommodation based on my disability”.*

*(Survivor 14, sarcoma)*


While strategic claiming of disability was seen as important for assertion of rights, several participants indicated that stereotypes of people with disabilities (especially hidden disabilities), getting services they did not deserve reinforced social stigma.


*I have a disability that people cannot see. I do have a handicapped placard because I know I’m a fall risk. The less time I spend walking around probably the better… This is another problem that I see with disabilities, is people are insensitive to that and one of their mottos is, ‘Oh, we’re so inclusive”. Well, if you’re inclusive, why do we have to fight for these things?*

*(Survivor 10, breast cancer)*


Acknowledging disability was not enough to counteract systemic and internalized ableism and guarantee fair treatment. A small group of participants was able to move from seeing disability as something that they had to acknowledge and live with, to affirming a disability identity and sense of community with other disabled people.

#### 3.4.4. Affirming

Survivors with greater exposure to the disability community were able to frame their experiences within the broader context of disability rights and affirm disability as a part of their identity. For example, survivor 21 spoke of her emerging understanding of the need to connect the disability and cancer communities.


*I do consider myself to be a person with a disability. I mean, my leg missing is pretty disabling… If I didn’t have a leg missing and I just had cancer, I don’t know if that would be true. Which is quite unfortunate for people in the cancer community because the disability community includes them as well… Apparently, having a disability, it’s like, you have to look or feel or act and cancer doesn’t make you feel that way. And so then also this negative stigmas and stereotypes about having a disability, like, like, no one wants to sit at the table with, like, the losers. Like that’s legit, how they don’t really want to be a part of it. But but we’re not losers… It’s really unfortunate, because there are laws in place that can protect people who have cancer, and [if they] would allow themselves to accept the help, they will be able to benefit greatly... So people who have disabilities just need to find some way to mesh with that word because things in place to protect you.*

*(Survivor 21, 2-time cancer survivor, childhood sarcoma, breast cancer)*


Survivor 9 highlighted how her pre-existing relationships within the disability community helped her frame her cancer experience and get community support to tackle challenges as they arose in her cancer journey.


*I’ve done a lot of work with the Americans with Disabilities Act, so I'm, I think, well informed about that. And I’ve also been a board member of [a disability rights organization], which is, they do a lot of work with people with disabilities… Their leadership, in fact, they were really, really very supportive and they would check on me all the time. And, and there was one woman in particular who said to me, ‘You know, you have a disability now, and don’t let anybody take your rights away, you know, away”. Because I had an instance at work when this first happened that, I was basically told in one of my reviews that, you know, I was no longer performing well because they were mistaking the fact that I was kind of having trouble speaking as a lack of confidence… It was hard for me personally because I had been with this company for so many years and was the senior person. I took it more as an insult that I even had to, like, go through all this explanation as to what was happening with me… And I think had I not had that relationship with [disability organization], [I wouldn’t] understand this reality of the disability community.*

*(Survivor 9, head and neck cancer)*


Similarly, based on her exposure to the disability rights movement, participant 13 was able to shift her lens of understanding from seeing her struggles and participation restrictions as a problem within herself to something that structural ableism has had a hand in creating and thus has a responsibility to eliminate.


*I think once I started to be more aware of… the history of the ADA and of like, the broader movements, social movements around accessibility… I think that, like the term disability and using the label disability was very valuable to me in terms of situating my experience inside something that isn’t just a problem to be solved... and is actually just a part of something larger, that hopefully society is figuring out how to deal with a little bit better. And so, it’s not only on me personally to fix myself. I think that was a relief.*

*(Survivor 13, breast cancer)*


Just as people who rejected disability identity could be seen as resisting the essentializing nature of disability stigma, the affirming group resisted the essentializing of disability as a major identity to integrate disability into their sense of self in ways that promoted their self-worth and collective conscience.

## 4. Discussion

This is one of the first studies to explore how cancer survivors experience and make sense of disability when confronted with the long-term physical, cognitive, sensory, and psychosocial effects of cancer and its treatment. The World Health Organization’s International Classification of Functioning, Disability, and Health (ICF) defines disability as impairments in body functions and structures, activity limitations, and/or participation restrictions that interact with environmental and personal factors to create the state of disability. Similarly, The Americans with Disabilities Act, the primary civil rights law in the United States prohibiting disability discrimination, defines a person with a disability as someone with a physical or mental impairment that substantially limits a major life activity. According to these definitions, cancer survivors experiencing long-term effects are considered people with disabilities, yet half of our participants did not identify as such. Participants used four approaches to negotiate their relationship to disability identity: rejecting, othering, acknowledging, and affirming. This indicates that disability identity is a complex social construct that shapes and is shaped by how people view themselves, their bodies, and their ways of interacting in the world [47]. These self-appraisals do not occur in isolation but are informed by a variety of individual and environmental factors. Individual factors may include symptom severity, visibility, and impact, as well as personal beliefs about disability, both positive and negative. Environmental factors include social attitudes, disability stigma [48], affiliation with disability communities [49], and access to resources.

The complex relationship that study participants had with disability is consistent with the existing literature on disability identity, which acknowledges its fluid and permeable nature [50,51]. As a fluid identity marker without rigid boundaries or clear definitions, disability is contextually and personally defined. There is no fixed or single way of being in the world and each cancer survivor must negotiate what works for them within and across a variety of social contexts. Some participants chose to distance themselves from a socially devalued status and thereby resisted the tendency of outsiders to essentialize disability and give it precedence over other attributes, interests, and qualities that a person possesses [4]. While this may be an adaptive strategy in some situations, research has shown that affirming or at least strategically using a disability identity is associated with higher collective and personal self-esteem [37] that can have protective effects on resisting pervasive structural ableism. The onus for addressing and dismantling structural ableism resides within the cancer care system and not within the individual survivor experiencing the long-term effects of cancer and its treatment.

Acknowledging and addressing structural ableism is important because pervasive experiences of ableism are associated with poorer health and well-being [10]. The impact of real and anticipated stigma has also been shown to decrease health and help-seeking behaviors and overall quality of life [9]. Real and anticipated stigma around disability was identified as one reason that participants eschewed a disability identity. Unfortunately, not talking about the disabling impacts of cancer and its treatments does not minimize the influence that they have on people’s post-treatment lives and ability to participate in roles and activities. Furthermore, providers’ reluctance to discuss the possibility of long-term disability may have the unintended consequence of reinforcing ableist stereotypes about disability as a fate worse than death [52]. Discomfort around disability may also limit frank discussions about the long-term effects of cancer and its treatments and contribute to survivors’ lack of preparedness for life post-treatment. This can have a ripple effect as lower levels of preparedness are associated with poorer symptom management [53]. Given the high symptom burden experienced by many cancer survivors, providing people with as many tools as possible to successfully transition to life post-treatment is essential for promoting cancer equity.

Access to interdisciplinary cancer rehabilitation is one such tool. Rehabilitation interventions can help improve outcomes related to pain, fatigue, balance, mobility, cognition, and anxiety and depression [54,55]. Unfortunately, fewer than 10% of cancer survivors who need cancer rehabilitation receive it [56]. Consistent with an extensive body of clinical and research literature, participants in this study reported significant physical, cognitive, sensory, and emotional impairments. Multiple impairments often intersected to pose significant barriers to survivors’ participation in major life roles, including their ability to care for themselves and others, work, volunteer, go to school, and socialize. It also limited their engagement in both active and quiet leisure activities. Inability to perform their activities adversely impacted people’s sense of competence, self-worth, and quality of life. In some cases, it also strained their social relationships and financial resources. In spite of the importance that participation has on people’s overall health and well-being [57], it receives limited attention in the current cancer continuum of care [58]. Indeed, there is a paucity of intervention research aimed at helping cancer survivors return to meaningful participation in their roles and responsibilities [59]. There is, however, a need to more fully support cancer survivors through the provision of evidence-informed interventions, including, for example, self-management [60,61] and mindfulness-based interventions [62,63] with an aim towards addressing participation restrictions, social connectedness, and amelioration of physical, cognitive, and psychosocial function. To reach the broadest group of survivors, interventions may be delivered in-person, hybrid or via mHealth tools.

While advocating for improved access to rehabilitation and self-management services and supports, it is important to recognize that these remain largely rooted in the biomedical model. The biomedical model’s focus on individual impairments fails to acknowledge the intersection of social, political, and economic factors that influence survivors’ disability experiences [64]. Cancer rehabilitation is not a cure for living with a chronic impairment. As in the example of survivor 1, despite advancements in prosthetics and adaptive equipment, living with an above-knee amputation will forever change the way she interacts with her environment and experiences the world. While there is no one *right way* to experience disability and disability identity is both fluid and porous, the shift from paternalistic medicine practices to acknowledging the material reality of living with a chronic impairment can be a critical step in ensuring more positive and equitable outcomes for long-term cancer survivors with disabilities.

Future research should continue to explore how structural ableism embedded in the cancer care system influences the services and supports available to people with pre-existing disabilities as well as people who acquire functional, cognitive, and psychosocial impairments as a result of their cancer. Future studies can build on the findings of this qualitative exploratory study to examine the causal and intersectional relationships of multiple marginalized groups related to cancer health equity and outcomes.

### Limitations

While this study had many strengths and yielded some novel findings, it was not without limitations. We drew from a convenience sample of people connected to cancer centers and cancer support organizations in a major urban center in the Midwestern United States. People in smaller, rural or less well-connected and resourced communities may report significantly different experiences in post-treatment life and disability identification. Our sample is also skewed towards white, middle-aged and female, which may again have limited the representativeness and transferability of the findings. We only targeted our recruitment to three groups of cancer survivors and within these groups, breast cancer survivors were significantly over-represented which may obscure issues of salience to other cancer survivor groups. The exploratory qualitative nature of these data does not enable us to examine the causal relationships between impairment severity, length of time in survivorship, cancer type and disability identity. Future qualitative and quantitative research should explore these relationships. Data were gathered cross-sectionally, so we only captured point-in-time data rather than the dynamic evolution and social construction of disability identification over time. This could be a productive area for future research.

## 5. Conclusions

According to contemporary and legal definitions, the roughly 40% cancer survivors who are living with long-term effects of cancer and its treatments are considered to be people with disabilities. Yet, only 50% of our participants identified themselves as a person with a disability in spite of pervasive physical, cognitive, and psychosocial long-term effects which resulted in significant participation restrictions in major life areas. Survivors exhibited a variety of approaches in navigating disability identity issues, including rejecting, othering, acknowledging, and affirming. Experienced, anticipated, and internalized stigma rooted in structural ableism may result in health inequities for cancer survivors. Cancer care must be intentionally anti-ableist. There is a need for frank discussions about the long-term disabling impact of cancer as well as the implementation of interventions to promote meaningful social participation amongst cancer survivors.

## Figures and Tables

**Table 1 ijerph-19-03112-t001:** List of Participants (*n* = 30).

Survivor	Cancer Type	Age (years)	Gender	Race	Last Cancer Treatment (year)
Survivor 1	Sarcoma; Breast	66	Female	White	2019
Survivor 2	Head and Neck	47	Male	White	2011
Survivor 3	Breast	53	Female	White	2014
Survivor 4	Head and Neck	38	Male	Unknown	2016
Survivor 5	Breast	53	Female	White	2021
Survivor 6	Head and Neck	70	Male	White	2012
Survivor 7	Breast	66	Female	White	2014
Survivor 8	Breast	58	Female	White	2016
Survivor 9	Head and Neck	61	Female	White	2017
Survivor 10	Breast	69	Female	White	2010
Survivor 11	Breast	69	Female	White	2010
Survivor 12	Head and Neck	72	Male	White	2013
Survivor 13	Breast	42	Female	White	2016
Survivor 14	Sarcoma	67	Female	White	2008
Survivor 15	Breast	66	Female	Multiracial	2020
Survivor 16	Sarcoma	22	Female	White	2020
Survivor 17	Breast	66	Female	White	2005
Survivor 18	Sarcoma	29	Female	White	2021
Survivor 19	Breast; Lung	69	Female	White	2003
Survivor 20	Breast	58	Female	White	2019
Survivor 21	Sarcoma; Breast	29	Female	Black	2019
Survivor 22	Breast	63	Female	White	2016
Survivor 23	Breast	62	Female	White	2015
Survivor 24	Breast	53	Female	White	2008
Survivor 25	Breast	48	Female	White	2015
Survivor 26	Head and Neck	59	Female	White	2016
Survivor 27	Breast	79	Female	White	2016
Survivor 28	Breast	69	Female	White	2011
Survivor 29	Breast	79	Female	White	2006
Survivor 30	Breast	59	Female	White	2006

**Table 2 ijerph-19-03112-t002:** Participant-identified long-term effects with representative quotations.

Body & Physical Changes	Representative Quotations
Decreased balance Decreased mobility Decreased endurance Decreased range of motion Dysphagia Dysarthria Limb loss Lymphedema Neuropathy Sensory loss—taste, smell Sleep Disturbances Vertigo	*I don’t eat in front of anyone. Because I do choke a lot when I eat. (Survivor 2, head and neck cancer)*
	*The neuropathy has given me difficulties with my balance. I’m extremely clumsy now. (Survivor 10, breast cancer)*
	*I have difficulty with the strength in my left shoulder and difficulty with range of motion. So certain items of clothing are harder to get off. Lifting is still a struggle and I still have neuropathy in that shoulder. Fatigue. (Survivor 18, sarcoma)*
Brain and Cognitive Changes	
Attention ‘Chemo Brain’ Memory changes Mental exhaustion Mental fuzziness Problem Solving Word Finding	*I have anomic aphasia, which affects my speech and my ability to process numbers and words, and my memory is affected, and I cannot learn new things readily because I don’t retain new information. (Survivor 8, breast cancer)*
	*I can’t retain things [crying]. Sorry it’s emotional. I have a hard time learning new skills. I’m an educated person and I had a functioning life, and I can’t function anymore. (Survivor 24, breast cancer)*
Thoughts and Feelings	
Anxiety Body image Depression Existential crisis Post-traumatic stress	*I just want to look normal. I don’t want to look like I have one breast. I don’t want to look disabled; I want to be normal…. Mentally I was probably at the worst point of my time because I couldn’t stop crying. I didn’t know what to do. I was really in a bad way. (Survivor 14, sarcoma)*
	*You realize your mortality and you’re becoming you know like sad like you’re going to miss out on things and so that’s depression. (Survivor 25, breast cancer)*

## Data Availability

The data presented in this study are available on request from the corresponding author. The data are not publicly available due to issues of privacy.

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
