# Peer review of "Cancer Survivors’ Disability Experiences and Identities: A Qualitative Exploration to Advance Cancer Equity"

_ijerph, 2022, doi:10.3390/ijerph19053112_

Round 1

Reviewer 1 Report

This is a very well-written manuscript in an area that is essential. Here are some suggestions to improve readability:

  1. The discussion about ICF in the Background section springs out of nowhere. It is a crucial discussion, but it needs to be tied to the content. As a reader, I was missing the fact about why we were discussing about the ICF. Was the interview designed based on the ICF, or were the results interpreted in light of the ICF? In the discussion section, the authors brought up ICF very briefly.
  2. Please provide a copy of the interview questions.
  3. On line 152, the authors reported monitoring updates to data saturation. Were 30 participants recruited because data saturation was reached? More information about data saturation is required. 
  4. Since member checking was not performed, how was response validation established? 
  5. Minor edits in spellings and grammar is needed throughout the paper. 

Author Response

Reviewer 1 - The discussion about the ICF in the Background sections springs out of nowhere. It is a crucial discussion, but it needs to be tied to the content. Asa reader, I was missing the fact about why we were discussing about the ICF. Was the interview designed based on the ICF, or were the results interpreted in light of the ICF? In the discussion section, the authors brought up ICF very briefly.

Response

The ICF is the dominant global framework for conceptualizing the dynamic interaction between body structures and function, activities and participation and environmental and personal factors and is rooted in a biopsychosocial approach. It provides a useful framework to anchor discussions of cancer symptoms and impact leading into conversations about the framing of disability in the US and around the world. It loosely structured aspects of both in the interview guide development, especially cancer’s long-term effects’ impact on doing (aka social participation) and well as our framing for the question on disability “According to some definitions, these things [impact on participation] would be considered a disability. Do you consider yourself to be a person with a disability?”

We have revised the section on the ICF to better establish the rationale for its inclusion as well as streamlining the description of the ICF activities and participation chapters in response to reviewer 1’s feedback.

Reviewer 1 - Please provide a copy of the interview guide

Response - A copy of the interview guide is included with the resubmission.

Reviewer 1 - On line 152, the authors reported monitoring data saturation. Were 30 participants recruited because data saturation was reached? More information on data saturation is required.

Response  - We originally used Guest et al.’s framework for defining sample size based on quantitative ‘rules of thumb’ for qualitative sample sizes. Broad themes of lack of preparedness, cancer symptoms experience, cancer’s impact on doing, , and the range of responses related to disability identity were firmly established by interview 21. We continued data collection to 30 for pragmatic and conceptual reasons. Pragmatic reasons were based on the substantial community interest and volunteers assertions that they are rarely asked about their long-term effects and that they really valued the opportunity to speak to those experiences so that other may learn from them. Therefore, given the minimal risk and burden to participants we sought an amendment to our IRB to increase our sample size and continued recruitment and data collection until all interested individuals for our community outreach efforts were screened, enrolled, and interviewed as appropriate. More conceptually from a social constructivist and reflexive  thematic analysis perspective, the notion of saturation is contested as people make and re-make meaning of fluid identity markers like survivorship and disability (Braun & Clarke, 2021).  Therefore, additional data collection beyond code and thematic saturation added depth and nuance to the understanding of cancer survivors’ disability experiences and identity.

We have added an abbreviated version of this rationale into the methods section of the manuscript.

Reviewer 1 - Since member checking was not performed, how was response validation established?

Response - Consistent with our epistemological and analytic frameworks of social constructionism, critical theory, and reflexive thematic analysis we used alternate strategies to ensure validity of findings as advocated by Creswell and Miller (2000) and Braun and Clark (2020). Specifically, we use peer debriefing by enlisting critical feedback data analysis and interpretation from cancer survivors and rehabilitation specialists with backgrounds in cancer rehabilitation. 

Reviewer 1 - Minor edits in spelling and grammar is needed throughout the paper.

Response - We have carefully edited and proofread the manuscript for spelling and grammar.

Reviewer 2 Report

General comments: This is a very interesting and well conducted study. What is the “cancer equity agenda”? Can that be more clearly described in the background? As written the main take home point seem to be that cancer survivors should accept the label of disability. I do not see that as being crucial to cancer survivorship. I could see it as being helpful for the cancer healthcare providers to keep in mind the importance of connecting cancer patients to the appropriate services such as rehabilitation as it is a poor link currently. I do think your findings are not unexpected. Depending on who you talk to, many people who have been diagnosed with cancer do not accept the label of survivor. The label of disabled is definitely not a label many people would like to take on. Yes it might be considered ableist, but not unexpected. What is interesting to me are the people who were willing to add that label. What influences that? Is there a level of disability or a type of disability that leads to that acceptance? Did they have pre-existing disabilities?

TITLE                                              

- The title clearly conveys the point of the article.

ABSTRACT

The abstract is accurate, concise, and clearly describes the study.

INTRODUCTION

- Previous pertinent literature cited and discussed- Yes, this is a very nice introduction to the problem and segue into the methods.

- Purpose/research hypotheses clearly stated- Yes

METHODS          

- The Study design was appropriate to achieve study objective

-In the participants section line121-123 please remove the parenthesis and connect the information into the sentence.

-I do have a question related to the self-identifying as a person with a disability. I am not clear how this was assessed. Was the term disability used? As I read I am confused if all the participants identified as having a disability how did your themes emerge (i.e. rejecting, othering)? Could you provide more detail about this for clarity?

-In the Data Collection section the 4 interview categories are quite varied from what is being reported in the paper. Is this paper only reporting on one of the 4 categories? If so please make that clear.

-In data collection the interviewers are described as having experience in disability and cancer rehab, but what about experience with qualitative methods/interviewing?                                                                                  

RESULTS                                                              

- Results clearly presented- There are some weaknesses included below.

-The list of participants table is very helpful. I am assuming the participants with both breast cancer and sarcoma are in fact people with metastasized breast cancer? That may need some clarifying, particularly if you are highlighting loss of limb for those with sarcoma. That information may need to be added to the table, along with type of treatment each person had since a lumpectomy may not have the same negative long term outcomes that chemo could. Stage of cancer and length of time since diagnosis might also help in terms of understanding the individual’s context. How many are long-term, how many are taking part in on-going palliative treatment, and how many have more recently completed treatment.

-Do you feel that you actually achieved saturation for those with head and neck cancer and sarcoma? I would believe that the breast cancer experience may be different form the head and neck (n=6), and sarcoma (n=2 only, 3 with breast cancer) participants.

-page 5 lines 184-187 highlights the issue mentioned above about how you tried to actually recruit those who identified as having a disability and yet that did not happen. It seems that how you screened was not appropriate. To a certain extent I would think that the person’s perception of their disability identity is the most important thing. This is the constructivist point of view, meaning what their beliefs are is there reality.

-Did age have any effect on the experiences? Unfortunately cancer often occurs at the same time that many other physical changes happen so it can be hard to decide what is causing what. Might be something to add to potential future questions.

-Many of your quotes come from those with breast cancer. This supports my question about the other two cancers being well represented.

DISCUSSION

- Previous pertinent literature critiqued and continuity between this present study and previous work is demonstrated. One thing that is not clear to me is how the intervention was customized. Is the same manual used for everyone, how can and was that customized as stated in the discussion?

- Limitations of study were noted and avenues for future research were provided.

CONCLUSIONS

- I question the first sentence of the conclusion stating that all cancer survivors are considered to be people with disabilities. I do not agree with this statement, it is too broad.

-I am not completely sure what the driving force behind this study is. Is it related to the lack of connection between cancer care and rehabilitation? If so make that point clearly somewhere (background, discussion). I think this is an important point and would help your argument. The link to structural ableism is interesting but almost seems like you are pushing an agenda that is leading to a sort of devaluing of the participant’s actual feelings. Is accepting a label of disabled really that important to an individual. I see that it certainly is in terms of policy (i.e. ADA in the work place) but is it the lack of accepting a label what is limiting access to care?

FORM, STYLE, AND SUBSTANCE

This is a well-organized and cohesive paper.

Author Response

Thank you for the thought provoking feedback. Our response to reviewer 2's comments are in the attached word document. We have taken the opportunity to critically reflect and strengthen work based on the reviewer's feedback.

Round 2

Reviewer 2 Report

Thank you for providing your revised paper. I believe your revisions have addressed all concerns.